# The Effect of Broccoli Sprout Extract on Seasonal Grass Pollen-Induced Allergic Rhinitis

**DOI:** 10.3390/nu13041337

**Published:** 2021-04-17

**Authors:** Joseph Yusin, Vivian Wang, Susanne M. Henning, Jieping Yang, Chi-Hong Tseng, Gail Thames, Irina Arnold, David Heber, Ru-Po Lee, Laura Sanavio, Yajing Pan, Tianyu Qin, Zhaoping Li

**Affiliations:** 1Department of Allergy and Immunology, Veterans Affairs Greater Los Angeles Healthcare System, Los Angeles, CA 90073, USA; vivian.wang@va.gov (V.W.); gthames@mednet.ucla.edu (G.T.); arnoldirina@gmail.com (I.A.); zli@mednet.ucla.edu (Z.L.); 2Department of Internal Medicine, University of California Los Angeles Medical Center, Los Angeles, CA 90095, USA; shenning@mednet.ucla.edu (S.M.H.); jiepingyang@mednet.ucla.edu (J.Y.); CTseng@mednet.ucla.edu (C.-H.T.); d.heber@mednet.ucla.edu (D.H.); RLee@gmail.com (R.-P.L.); lsanavio@ucla.edu (L.S.); panyajing152@sina.com (Y.P.); qintianyu@bucm.edu.cn (T.Q.)

**Keywords:** glutathione transferase, sulforaphane, broccoli sprout, allergic rhinitis, antioxidant, nasal corticosteroid, peak nasal inspiratory flow, total nasal symptom score, T2 cytokines, allergen extract

## Abstract

Patients exposed to pollutants are more likely to suffer from allergic rhinitis and may benefit from antioxidant treatment. Our study determined if patients diagnosed with grass-induced allergic rhinitis could benefit from broccoli sprout extract (BSE) supplementation. In total, 47 patients were confirmed with grass-induced allergic rhinitis and randomized to one of four groups: group 1 (nasal steroid spray + BSE), group 2 (nasal steroid spray + placebo tablet), group 3 (saline nasal spray + BSE) and group 4 (saline nasal spray + placebo tablet). Peak Nasal Inspiratory Flow (PNIF), Total Nasal Symptoms Scores (TNSS) and nasal mucus cytokine levels were analyzed in samples collected before and after the 3-week intervention. Comparing before and after the intervention, PNIF improved significantly when comparing Groups 1 and 2, vs. placebo, at various time points (*p* ≤ 0.05 at 5, 15, 60 and 240 min) following nasal challenge, while TNSS was only statistically significant at 5 (*p* = 0.03), 15 (*p* = 0.057) and 30 (*p* = 0.05) minutes. There were no statistically significant differences in various cytokine markers before and after the intervention. Combining nasal corticosteroid with BSE led to the most significant improvement in objective measures.

## 1. Introduction

Oxidant stress from pollution could be a contributing factor in the growing incidence of atopic disorders in urban areas [1]. This observation is supported by studies involving animals exposed to pollutants, including diesel exhaust particles. After exposure, these animals produced increased numbers of IL-4, a cytokine product of TH2 cells, and IgE antibodies that attach to mast cells, causing degranulation and the eventual allergic reaction [2,3,4,5,6]. Human studies involving exposure of nasal mucosa to diesel exhaust particles led to increases in nasal mucus IgE [7,8,9].

In vitro studies support the concept that pollution contributes to the allergic reaction. NADPH oxidase located within pollen grains can generate reactive oxygen species (ROS) [10], leading to increases in specific IgE [11]. In addition, diesel exhaust particles (DEP) interact with the eotaxin gene causing increase production of eotaxin [12], an eosinophil chemoattractant that is vital for the propagation of allergic reactions.

These studies indicate that antioxidants could benefit patients suffering from atopic disorders, including allergic rhinitis. Serum IgE levels of mice exposed to an allergen plus diesel exhaust particles decreased after exposure to thiol antioxidants [13]. However, several early clinical trials treating atopic asthma with antioxidants failed to show benefits [14,15,16]. It is possible that only select groups of patients would improve on antioxidant therapy.

A significant number of patients diagnosed with atopic asthma are more likely to have null genotypes for glutathione transferase (GST), which is essential for the production of phase II enzymes that act as antioxidants [17]. These individuals tend to have an exaggerated T2 response to diesel exhaust [18]. Asthmatic children with the GSTM null genotype showed superior improvements in lung function after therapy with the antioxidants vitamin E and C, compared to GSTM1-positive asthmatic children [19,20].

Sulforaphane (SFN) was originally isolated from broccoli as a byproduct [21] formed by enzyme myrosinase activity on glucoraphanin [22]. SFN has antioxidant activity through activation of nuclear factor E2-related factor 2 (Nrf2), which leads to stimulation of the GST Phase II gene products [23]. Administration of SFN increased expression of phase II enzymes and decreased DEP-induced production of proinflammatory cytokines IL8, GM CSF and IL1b in mice bronchial epithelial cells [24]. These findings were reproduced in a human model that showed administration of broccoli sprout homogenate increased in nasal epithelial cell phase II enzyme expression [25].

This study’s primary objective was to compare clinical measurements for allergic rhinitis following grass challenge before and after a 3-week course of a sulforaphane supplement. The secondary objectives were to compare biomarkers before and following the 3-week course of BSE supplementation and to conduct an exploratory genetic analysis of the 3 glutathione S-transferase (GST) genes (GSTM1, GSTT1 and GSTP1).

## 2. Methods

### 2.1. Study Design

This was a double-blind, randomized, placebo-controlled trial to compare clinical measurements for allergic rhinitis following Timothy (Phleum Pretense), Bermuda (Cynodon Dactylon) or Johnson (Sorghum Halepense) grass allergen challenge before and after initiation of a 3-week SFN vs. placebo supplementation.

### 2.2. Participants

Participants included non-smoking male and female US veterans followed at the VA Greater Los Angeles Healthcare System. Each participant had a history of symptoms consistent with seasonal allergic rhinitis (rhinorrhea, nasal congestion, postnasal drainage, sneezing, and/or nasal or eye pruritis) to grass pollen (symptoms during summer months, June through August), for a minimum of two consecutive seasons. Participants were excluded if they experienced symptoms most of the year (9 or more months). These participants were not taking any medications for allergic rhinitis at the time of enrollment.

Key exclusion criteria included: (1) symptoms consistent with allergic rhinitis for 9 or more months of the year; (2) current use of antihistamines, leukotriene antagonists, or nasal steroid sprays; (3) uncontrolled or serious medical or surgical condition that, in the opinion of the investigators, could affect the subject’s safety and/or interfere with the study assessments; (4) history of anaphylaxis to environmental allergens or an unknown trigger; (5) history of broccoli allergy; (6) recent upper respiratory infection (less than 4 weeks prior to study) or other active infection; (7) currently receiving allergy immunotherapy; (8) history of rhinitis exacerbation within the past 2 weeks; (9) use of non-selective Beta-Blocker; (10) inability to give written informed consent, history or evidence of non-stable cognitive capacity within less than 1 year (i.e., Alzheimer’s disease, dementia, bipolar disorder) that, in the opinion of the investigator(s), could affect the subject’s safety and/or interfere with the study assessments; (11) pregnancy; and (12) uncontrolled asthma or a forced expiratory volume in 1s (fev1) < 70% predicted at screening.

After consent, subjects underwent skin testing to the following grass allergen extracts: P. Phleum (10,000 bu/cc), C. Cynodon (10,000 bu/cc), or S. Sorghum (1:100 *w*/*v*). A positive skin test was defined as a wheal of 3 mm or greater than the negative control. A positive result indicated confirmed allergic rhinitis to grass pollen. The subjects proceeded with the study (Figure 1).

### 2.3. Study Objectives

The primary objective of the study was to compare clinical measurements for allergic rhinitis following Timothy, Bermuda or Johnson grass allergen challenge before and after initiation of a 3-week BSE supplement (glucoraphanin plus myrosinase, which together forms sulforaphane) Clinical measurements were obtained using the total nasal symptoms score (TNSS) and peak nasal inspiratory flow (PNIF). The secondary objective was to compare biomarker measurements from nasal mucus following the grass allergen challenge before and after initiation of BSE supplementation. During nasal challenges, the following cytokines were measured: IL4, IL5, IL13, IL6, IL8 and IL1b. An exploratory genetic analysis of the 3 GST genes, GSTM1, GSTT1 and GSTP1, for each participant was performed to determine the effect of antioxidant genetic polymorphisms in the response to BSE supplementation on grass-induced allergic rhinitis.

### 2.4. Study Products Used during the Study Included

Broccoli sprout extract: Avmacol (Nutramax, Lancaster, SC, USA) is a blend of broccoli seed and sprout extracts contains the 2 essential ingredients needed for sulforaphane production: glucoraphanin and the myrosinase enzyme. Two375 mg tablet contain ~30–35 umoles sulforaphane per recommended serving. Placebo tablets contained water, HPMC, triactin, mineral oil, sodium lauryl sulfate, titanium dioxide. The appearance of the placebo was identical with Avmacol. Subjects were given 4 tablets to take with their evening meal. Following baseline nasal challenge, subjects were randomized into one of 4 groups:nasal corticosteroid + placebo tabletsaline nasal spray + placebo tabletnasal corticosteroid + BSEsaline nasal spray + BSE

### 2.5. Study Procedures

The challenge procedure and measurements were based on a tested method developed by Guy Scadding [26].

Following confirmation of a positive skin test to the grass allergen, participants underwent an up-dosing grass allergen nasal provocation test corresponding to the identical grass allergen determined from skin testing. Subjects experiencing a positive nasal provocation test proceeded to visit 2. A positive nasal provocation was defined as a TNSS over 7, or a PNIF reduction of over 50% from baseline. Blood was obtained from each qualifying participant to test for serum GSTM1 polymorphism.

Participants returned for the second visit 14 days following visit 1 and underwent the grass challenge to the effective dose, determined during the up-dosing challenge visit. Following the challenge, patients were monitored over a 4 h period, in which PNIF and TNSS measurements were performed hourly. Nasal effluent was collected using polyurethane sponges placed into both nostrils. The nasal fluid was analyzed for cytokines. Following visit 2, the participants were randomized into 1 of 4 groups (1. nasal corticosteroid + BSE (*n* = 15), i.e., steroid + BSE arm; 2. nasal corticosteroid + placebo tablet (*n* = 9), i.e., steroid arm; 3. nasal saline spray + BSE (*n* = 12), i.e., BSE arm; and 4. nasal saline spray + placebo tablet (*n* = 8), i.e., placebo arm. Participants returned for visit 3 (35 days from visit 1) after 3 weeks of intervention. They underwent a repeat challenge similar to visit 2.

### 2.6. Nasal Allergen Challeng

Grass pollen extract was used based on skin test results. During the up-dosing nasal challenge, the TNSS and PNIF were recorded immediately before each subsequent dose. For Bermuda Grass (C. Dactylon): 30, 300, 1000, 3000 and 10,000 bu/mL; for Timothy Grass (P. Pretense: 300, 1,000, 3,000, 10,000 and 100,000 bu/mL; for Johnson Grass (S. Halepense): 1:200,000, 1:20,000, 1:2000, 1:200 and 1:20 *w*/*v*. Participants received three sprays equivalent to 100 µL for each concentration into each nostril using an atomizer device, applied every 10 min until maximal concentration or a TNSS score of 7 or PNIF reduction by >50% from baseline reading was achieved. Prior to subsequent challenges at visits 2 and 3, nasal lavage (SinusRinse) was performed. A baseline PNIF and TNSS was recorded 30 min following nasal lavage for visits 2 and 3. Nasal challenge with 100 µL of the final dose achieved at visit 1 was applied into each nostril. The PNIF and TNSS was recorded following the challenge at 5, 15 and 30 min and 1 h, and then at each hour for 4 h. During these intervals, nasal sponges (Plastocell & Co Schifferstadt, Germany.) were applied into each nostril onto the nasal mucosa, beyond the nasal vestibule and alongside the inferior turbinate with the use of a crock forceps (Phoenix surgical instruments, Harlow, UK) and a Thuddicum’s nasal speculum (Phoenix surgical instruments, Harlow, UK). After two minutes of placement, the sponges were removed and then added to 2 mL centrifuge tubes with indwelling 0.22 µm cellulose acetate filters (Costar Spin-X, Corning, NY, USA) and underwent centrifuging. The liberated fluid was then frozen at −80° until analysis.

### 2.7. Cytokine Analysis

IL-1β, IL-4, IL-5, IL-6, IL-8 and IL-13 were analyzed in nasal fluid using the human cytokine panel HCYTOMAG-60K-06 (EMD Millipore, Billerica, MA, USA) with the Luminex MagPix^®^ analyzer (Luminex, Austin, TX, USA).

### 2.8. Genotyping GSTP1, GSTM1, GSTP1

DNA was extracted from buffy coat using the GenElute Mammalian Genomic DNA Miniprep Kit (Sigma–Aldrich, St. Louis, MO, USA). GSTP1 single nucleotide substitution (A- > G) was determined using the TaqMan Applied Biosystems SNP genotyping assay for GSTP1 (rs1695) C_3237198_20 (Thermo Fisher Scientific, Pittsburgh, PA, USA). GSTM1 and GSTT1 copy number was determined using the SNP genotyping assays Hs02575461_cn and Hs00010004_cn with human RnaseP as TaqMan copy number reference and genotyping master mix from Applied Biosystems (Thermo Fisher Scientific, Waltham, MA, USA). All assays were performed on a QuantStudio 5 real-time PCR system (Life Technologies, Grand Island, NY, USA).

### 2.9. Statistical Analysis

Summary statistics (mean, standard deviation and frequency distribution) were generated for baseline demographic and genotyping information. The primary endpoint of the study is peak nasal inspiratory flow (PNIF). Area under curve (AUC) was calculated for PNIF before and after intervention (day 14 and day 35). Linear model was used to evaluate the impact of BSE supplementation on the AUC on day 35, adjusting for the use of nasal corticosteroid and AUC on day 14. GST polymorphism effect, interaction effects between BSE supplementation and nasal corticosteroid and interaction effects between BSE supplementation and GST polymorphism were evaluated. Similarly, linear models were used to evaluate the effects of BSE supplementation and nasal corticosteroid on TNSS and inflammatory markers, based on linear models. All analyses were performed using R software (www.r-project.orgR version 3.5.1 Copyright (C) 2018 The R Foundation for Statistical Computing). All tests are two-sided, and a *p*-value < 0.05 is considered statistically significant.

## 3. Results

### 3.1. Baseline Characteristics

A total of 75 subjects proceeded with up-dose challenge, and a total of 47 were randomized after having a positive challenge. Out of the 47 participants, 1 participant withdrew from the study from group 3 (BSE) due to mild gastrointestinal complaints, and 1 participant from the same group failed to produce an adequate amount of mucus for analysis, leaving a total of 45 participants for analysis. Most subjects were classified as Caucasian or African American males, with the average age range of 44 to 59 years (Table 1).

### 3.2. Peak Nasal Inspiratory Flow (PNIF) and Total Nasal Symptom Score (TNSS)

PNIF and TNSS values were obtained at various time points and area under the curve was calculated and compared between baseline and completion of the intervention. Using the linear model, we determined that BSE supplementation significantly improved PNIF (*p* = 0.04). Using the post hoc test to compare the difference between groups, we found that area under the curve of changes in PNIF measurements during the intervention (day 35–day 14) was significantly increased with the combination of Steroid + BSE (group 1) compared to placebo (group 4) (*p* = 0.013) (Figure 2A). We also observed a trend of increased PNIF after BSE supplementation with BSE (group 3) compared to placebo (group 4) (*p* = 0.12). The difference in PNIF measurements between completion and baseline visits when compared to placebo (group 4) showed statistical significance at 5 (*p* = 0.029), 15 (*p* = 0.018), 60 (*p* = 0.02), 180 (*p* = 0.04) and 240 (*p* = 0.04) minutes following the nasal challenge in the Steroid + BSE group (group 1) and statistical significance at 60 min (*p* = 0.02) in the Steroid group (group 2) and BSE group (group 3) (*p* = 0.01) (Figure 2B). The difference in TNSS between completion and baseline visits when compared to placebo (group 4) showed statistical significance at 5 (*p* = 0.03), 15 (*p* = 0.057) and 30 (*p* = 0.05) minutes following the nasal challenge in the Steroid + BSE group (group 1) (Figure 3B). The area under the curve failed to show statistical significance (Figure 3A).

### 3.3. Inflammatory Markers

#### 3.3.1. Nasal Mucus Cytokine Levels

The mean differences between cytokine levels contained in mucus before and following the intervention was calculated for various time points following the nasal allergen challenge. Although decreased levels in T2 cytokines (IL-4, IL-5, IL-13) at various time points after the challenge were larger for the steroid +BSE group (group 1) compared to the placebo (group 4), there was no statistical significance (*p* > 0.05). There were no differences in IL-1β, IL-6 or IL-8 among the 4 groups (Figure 4). Areas under the curve were not statistically different among all cytokine measurements.

#### 3.3.2. GST Polymorphism

Glutathione-S-transferase gene copy number variance and polymorphism for GSTT1, GSTM1 and GSTP1 were analyzed by extracting DNA from buffy coat. GSTM1, GSTT1 and GSTP1 genotype distributions are listed in Table 2. GSTM1 null, GSTT1 null and GSTP1 single nucleotid polymorphis (Ile→Val) have been found to exhibit lower enzyme activity [27]. Adjusting for GST polymorphisms did not show any significant differences in PNIF, TNSS and cytokine concentrations among the four groups studied.

## 4. Discussion

Recent investigations indicate a role of antioxidants/supplements in the treatment of asthma and allergic rhinitis [28,29]. To further investigate this benefit for allergic rhinitis, we performed a double-blind, placebo-controlled study using a sulforaphane-based antioxidant We chose validated Peak Nasal Inspiratory Flow (PNIF) and Total Nasal Symptom Score (TNSS) to measure the four cardinal features of allergic rhinitis: nasal congestion, rhinorrhea, sneezing and ocular/nasal itching [30].

### 4.1. Clinical Benefit

The addition of an antioxidant to conventional nasal corticosteroid therapy for allergic rhinitis was proven beneficial in an open-label study [29]. Our study confirms these findings through analysis of PNIF. The area under the curve was statistically significant throughout the 4 h time period following the grass allergen challenge, indicating an objective improvement in nasal congestion. Compared to placebo, the group taking the combined oral sulforaphane and intranasal corticosteroid initially showed statistically significant improvement in the TNSS, though the improvement was not sustained after 30 min following the nasal challenge. These observations imply that BSE may interfere with the early phase reaction involving release of inflammatory mediators from mast cells, though it failed to prevent the late phase reaction, which involves recruitment of eosinophils and other atopic mediators [31]. The area under the curve graph for TNSS indicates a better response for patients taking nasal corticosteroid, alone, vs. in combination with oral sulforaphane. The subjective manner of the TNSS could explain this discrepancy, along with the small number of patients participating in the study. Future large studies should be conducted to confirm the clinical outcome.

### 4.2. Cytokine Levels

Cytokine levels were lower in nasal mucus from the intervention groups, though statistical significance was not obtained. Although patients were monitored for 4 h following the nasal challenge, the original study utilizing the same grass challenge model noted the most significant changes in cytokine levels after 4 h following the nasal challenge [26]. This finding was replicated in other studies looking at the benefit of nasal corticosteroids, alone, in allergic rhinitis, with the greatest effect on Th2 and other inflammatory cytokine levels after 4 h following the nasal challenge [32,33]. These studies highlight the importance of checking tryptase and eosinophils, which tend to peak at 1–2 h following nasal allergen challenge. We were unable to obtain enough mucus to analyze these inflammatory markers.

### 4.3. Genetic Analysis

A portion of our research endeavor was focused on determining whether GST polymorphisms predicted the outcome with the use of sulforaphane. Patients with GST null genotypes should experience greater benefit, since sulforaphane could supplement antioxidant support through inducing the phase II expression lacking in these individuals. GST null polymorphisms have been associated with a higher incidence of inflammatory conditions [34,35], including atopic asthma [36]. Unlike a prior study showing a benefit to GST null asthmatics with the addition of antioxidants [20], our study failed to show a correlation between patients with GST null genotypes and benefit from the sulforaphane supplementation (*p* > 0.1). Failure to show correlation could be attributed to low subject numbers. Larger number of patients are needed in future studies to determine if there are any significant associations between GST null genotypes and sulforaphane response.

### 4.4. Study Limitations

The major limitation of this study is the small sample size. In addition, there were issues with collecting adequate amounts of mucus following the challenges that prevented analysis of other mediators that are known to peak within the first few hours following allergen nasal challenges. Since our patients’ visits did not exceed 4 h following the challenge, we may have missed cytokine peaks, which could impact the determination of a significant response to sulforaphane following the grass allergen challenge.

## 5. Conclusions

Our study shows both subjective and objective improvement with the combination of nasal corticosteroids and sulforaphane supplementation, when compared to placebo tablets and nasal saline. These findings support a role of antioxidant supplementation in treating patients diagnosed with allergic rhinitis residing in urban areas with high pollutant levels. Larger studies are needed determine if oral sulforaphane is truly beneficial in treating allergic rhinitis.

## Figures and Tables

**Figure 1 nutrients-13-01337-f001:**
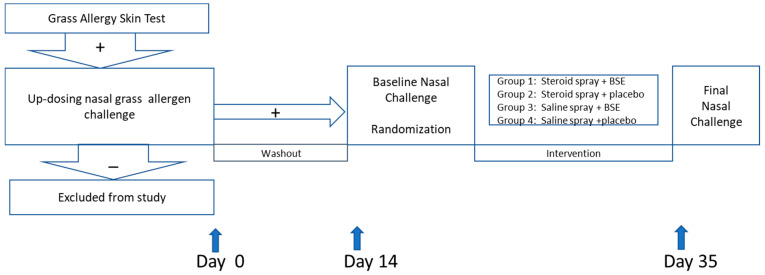
Overview of Study Protocol. Subjects were recruited based on clinical symptoms consistent with allergic rhinitis to grass pollen and verified with skin testing. If subject tested positive to grass pollen, they proceeded to an up-dosing nasal challenge for confirmation. Subjects who had positive challenge test returned in 2 weeks for the baseline nasal challenge and randomization, then returned 3 weeks after visit 2 for final nasal challenge.

**Figure 2 nutrients-13-01337-f002:**
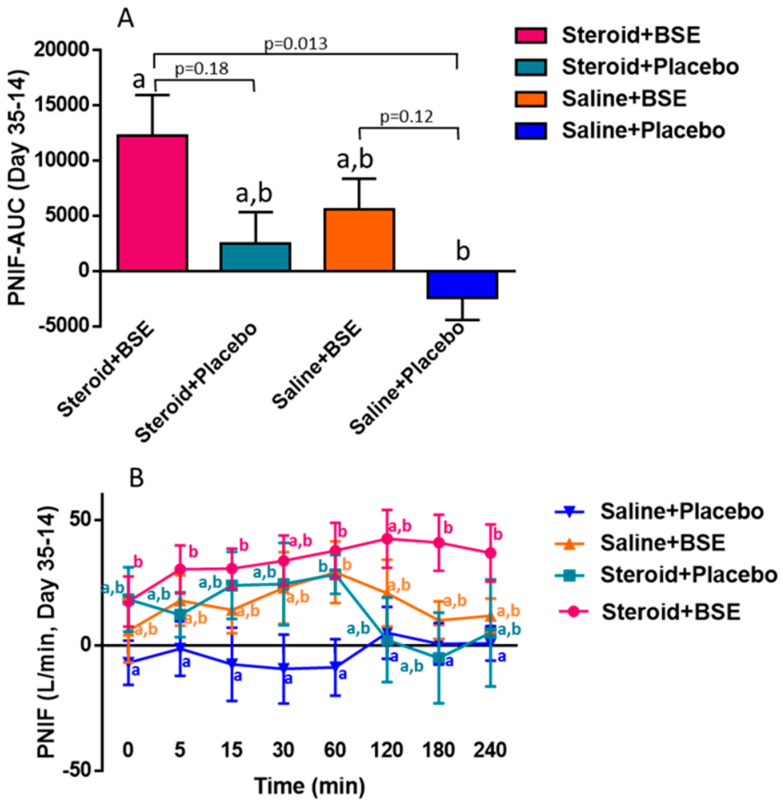
Differences between before intervention (day 14) and completion of intervention (day 35) for PNIF. AUC was calculated for PNIF among the 4 groups, showing a mean of 0.3362 (sd 0.0026) in group 1, 0.1213 (0.0607) in group 2, 0.0636 (0.0485) in group 3 and −0.0538 (0.0453) in group 4 (**A**). Changes over time are displayed (**B**). Differences in letters indicate statistical significance.

**Figure 3 nutrients-13-01337-f003:**
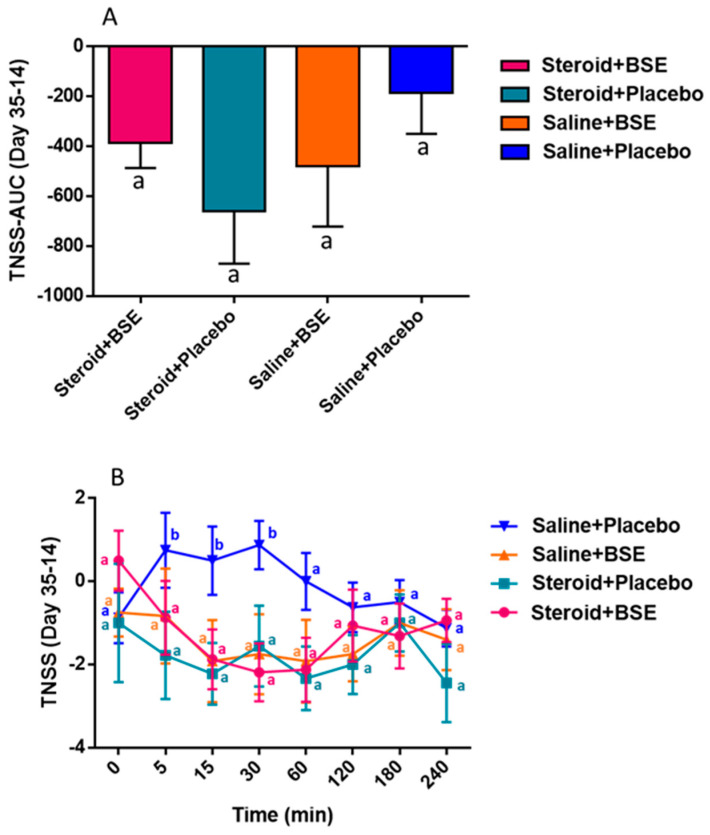
Differences between before intervention (day 14) and after intervention (day 35) for total nasal symptoms score (TNSS) were recorded. AUC was calculated for TNSS among the 4 groups, showing a mean of −0.5277 (sd 0.1048) in group 1, −0.8359 (sd. 0.3903) in group 2, −0.5484 (sd. 0.0014) in group 3 and −0.181 (0.0846) in group 4 (**A**). Changes over time are displayed (**B**) Differences in letters indicate statistical significance.

**Figure 4 nutrients-13-01337-f004:**
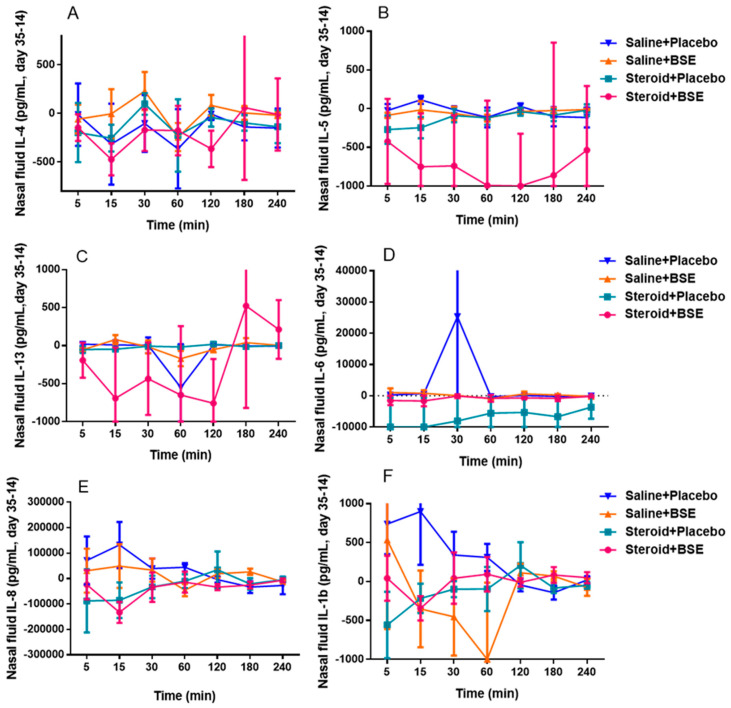
Differences between day 14 (before intervention) and day 35 (completion of intervention) for nasal mucus cytokine levels, divided into T2 cytokines (**A**–**C**) and other cytokines (**D**–**F**).

**Table 1 nutrients-13-01337-t001:** Subject description for each group, including: age, gender and race.

	Group 1: Nasal Steroid + BSE	Group 2: Nasal Steroid + Placebo	Group 3: Saline Spray + BSE	Group 4: Saline Spray + Placebo
Number of patients	16	9	14 *	8
Age (years), Mean (SD)	45.1 (12.2)	58.6 (13.5)	48.8 (15.9)	44.5 (18.5)
Gender (male)	75%	78%	83%	63%
Race, No. (%)				
Asian	3 (19%)	1 (12%)	0 (0%)	0 (0%)
Pacific Islander	2 (13%)	0 (0%)	0 (0%)	1 (12%)
Black	4 (25%)	4 (50%)	5 (42%)	4 (50%)
White	7 (44%)	3 (38%)	7 (58%)	3 (38%)

* 1 patient did not produce enough mucus for analysis, and 1 patient dropped out of study; thus, total analyzed was *n* = 12.

**Table 2 nutrients-13-01337-t002:** Breakdown of GST polymorphisms for each group. GST = glutathione S-transferase. Gene families divided into M1 null, M1, 2, 3 or 4 vs. T1 null, T1, 2 vs. P1G/G, A/G or A/A.

	Group 1 (*n* = 16)	Group 2 (*n* = 9)	Group 3 (*n* = 11)	Group 4 (*n* = 8)
GSTM1 null	9	3	4	2
GSTM1 (2,3,4)	7	6	7	6
% null	56	33	36	25
GSTT1 null	5	4	1	2
GSTT1 (1,2)	11	5	10	6
% null	31	44	9	25
GSTP1 G/G *	8	4	5	3
GSTP1 A/G *	5	4	4	2
GSTP1 A/A	3	1	2	3
% low activity	81	89	82	63

* GSTP1 G/G and A/G have low activity.

## Data Availability

Not applicable.

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
