# Peer review of "The Effect of Broccoli Sprout Extract on Seasonal Grass Pollen-Induced Allergic Rhinitis"

_nutrients, 2021, doi:10.3390/nu13041337_

Round 1

Reviewer 1 Report

It's an interesting and well conducted study with a correct study design but a part of methods is lacking  and results and discussion can be improved.

ABSTRACT : At the end you should outline that this therapy seems to be useful as objective parameters improve but data on TNSS are not so conclusive.

Pag 2 At the beginnnig modify the first sentence : is not a clear english Delete paragraph "we originally investigated broccoli...." as the reduction of white blood cell counts is not important in the pathogenesis of allergic rhynitis and allergic diseases

The next paragraph should be modified in: The primary objective of the study was to compare .............. (that when combined forms sulforaphane), because the data of patients enrolled are specified in methods.

METHODS The diagnosis of allergic rhinits is based on clinical history supported by SPT ( first level test ) or IgE specific dosage (second level). If you used SPT you should specify the extract used and how you consider positivities. Then did you use molecular based allergy diagnostics to distinguish genuine versus cross-reactive sensitization in  poly-sensitized  patients?  Did you include in your study patients poly-sensitized to  different pollens (birch, olive grass) or only monosensitized to grass pollens? You should specify.

 ARIA document classified rhynitis  in persistent and intermittent ; when you exclude from your study perennial rhinitis do you mean patients sensitized to perennial allergen? You should specify

Why did you record PNIF and TNSS following challenge at 5 minutes? In literature generally the record was done at 15, 30  60 minutes

Pag 3 : the dose and time of tablets recommended is not clear : 2 once a day? Specify

RESULTS: 3.2 day14-35 instead of 35-14

At the end of the pag 5 ERROR: fig 1d instead of 1c and 1c instead of 1d

 FIGURES. Fig 1a is written at the end of page and not in next page. The same error for Fig 1c. In Fig1a. You forgot to put the significant difference  (p=0,013) between group 1 and 4. Is  clearer to put a legend to each figure ( 1a,1b,1c, 1d) and control exactly the text (In Fig 1c the data of group 2 and 3 are reversed)

In FiG Fig 1b and 1d specify what does it mean a and b

The results obtained with TNSS and PNIF are different. I was expecially surprised  of the efficacy expressed in Fig 1 a and 1c using steroids ( data completely different!!).  As PNIF and TNSS are two different methods  to evaluate allergic rhinitis that inversely correlate, didn't you try to correlate data of obtained with theses  two tools? This should consolidate your results and the usefulness of PNIF  as objective control parameter for rhinitis symptoms. 

DISCUSSION In the first paragraph you mention various investigations on the role of antioxidants in the treatment of allergic rhinitis but the reference was only for the one DBPC study . As this is a new topic insert other references

4.2 You should try to explain why the efficacy of the new therapy is significant  only for 30 minutes after nasal challenge. This could be an inportant limit in its future use in clinical practice. 

CONCLUSION You should modify the first sentences. The subjective benefit of this therapy is not so evident!

REFERENCES Control orthograpy error and punctuation  in 7, 10, 15 24,26,28,32,37

Author Response

ABSTRACT: 

Point 1:  At the end you should outline that this therapy seems to be useful as objective parameters improve but data on TNSS are not so conclusive

Response 1:   Last sentence in the abstract now mentions failure to prove subjective clinical benefit based on TNSS following 30 minutes post challenge.     

Page 1: line 28-30. 

Point 2: Page 2 At the beginnnig modify the first sentence : is not a clear english Delete paragraph "we originally investigated broccoli...." as the reduction of white blood cell counts is not important in the pathogenesis of allergic rhynitis and allergic diseases

Response 2: Paragraph was deleted.   

Page 2: line 85-88.

Point 3:  The next paragraph should be modified in: The primary objective of the study was to compare .............. (that when combined forms sulforaphane), because the data of patients enrolled are specified in methods

Response 3:  specific data on patients enrolled was omitted from paragraph.

Page 2: line 89-90.  

METHODS

Point 1:   The diagnosis of allergic rhinits is based on clinical history supported by SPT ( first level test ) or IgE specific dosage (second level). If you used SPT you should specify the extract used and how you consider positivities. Then did you use molecular based allergy diagnostics to distinguish genuine versus cross-reactive sensitization in  poly-sensitized  patients?  Did you include in your study patients poly-sensitized to  different pollens (birch, olive grass) or only monosensitized to grass pollens? You should specify

Response 1:

Subjects had a history of at least 2 seasons of symptoms consistent with allergic rhinitis during summer season, when grass pollen is highest in our region.  We excluded patients who had symptoms consistent with perennial allergic rhinitis (symptoms occurring throughout the majority of the year).    After consent, subjects underwent skin testing if not performed within the last year.   Skin tests included grass allergen extracts (10,000 BU/cc for p. phleum and c. cynodon, or 1:100 w/v for s. sorghum).  In addition, the skin test included  other seasonal (weeds,trees) and perennial (dust mite,mold) allergens.  A positive skin test to the other allergens did not exclude patient from the study as long as the patient had symptoms consistent with grass allergy (symptoms during June to August).     

Page 3: line 107-108, 120, 123-127.  

A positive skin test was determined to be 3 mm or greater than negative control. 

Page 3: line 125.    

We did not perform molecular based allergy diagnostics though based the diagnoses of seasonal allergic rhinitis to grass on their clinical pattern (symptoms during summer months) combined the positive grass skin test.    

Point 2:   ARIA document classified rhinitis  in persistent and intermittent ; when you exclude from your study perennial rhinitis do you mean patients sensitized to perennial allergen? You should specify

Response 2:

 Perennial Rhinitis in our study was defined as patients with symptoms c/w allergic rhinitis for the majority of the year (9 or more months).   

Page 3: line 107-108.  

Point 3:   Why did you record PNIF and TNSS following challenge at 5 minutes? In literature generally the record was done at 15, 30  60 minutes

Response 3:   Following the challenge, when following PNIF and TNSS measurements, sponges were placed within the nose in order to collect mucus.   We decided to begin measurements at 5 minutes in order to collect adequate amounts of mucus for cytokine analysis.  

Point 4:  Page 3 : the dose and time of tablets recommended is not clear : 2 once a day? Specify

Response 4:   This has been corrected.  Each Avmacol tablet is 375mg.  Patients will take 4 of the 375mg tablets with their evening meal. 

Page 4: line 147, 160-161.    

RESULTS

Point 1: 

3.2  day14-35 instead of 35-14

Response 1:

Day 35-day 14 was kept, since in figure 1 the AUC difference is positive, so PNIF on day 35 was larger than day 14.  If it is day 14-35 it would be negative.  

Page 7: line 172

Point 2:  

At the end of the page 5 ERROR: fig 1d instead of 1c and 1c instead of 1d

Response 2:  

Actualy further review, it does look like figure 1c is the trend vs. 1d is the area under the curve. We have corrected the figures and relabelled.       

Point 3:

FIGURES. Fig 1a is written at the end of page and not in next page. The same error for Fig 1c. In Fig1a. You forgot to put the significant difference  (p=0,013) between group 1 and 4. Is  clearer to put a legend to each figure ( 1a,1b,1c, 1d) and control exactly the text (In Fig 1c the data of group 2 and 3 are reversed)

Response 3

The figures are now aligned with the title. We added the significance difference between group 1 and 4 to figure 1A. We revised the figures and now have two figures. Figure 2 presents PNIF and Figure 3 TNSS.

Point 4:

In FiG Fig 1b and 1d specify what does it mean a and b

Response 4:

As mentioned in the footnote, significant differences are indicated by superscript letters. Labeled means without a common letter differ,p<0.05. We have added this information to the figure legends.

Point 5:

The results obtained with TNSS and PNIF are different. I was expecially surprised  of the efficacy expressed in Fig 1 a and 1c using steroids ( data completely different!!).  As PNIF and TNSS are two different methods  to evaluate allergic rhinitis that inversely correlate, didn't you try to correlate data of obtained with theses  two tools? This should consolidate your results and the usefulness of PNIF  as objective control parameter for rhinitis symptoms

Response 5:

Differences between the 2 groups are explained in the discussion

Page 14: line 417-421.   

DISCUSSION

Point 1:

In the first paragraph you mention various investigations on the role of antioxidants in the treatment of allergic rhinitis but the reference was only for the one DBPC study . As this is a new topic insert other references

Response 1:

This article was replaced with another.  The original article mentioned this concept in the discussion though the current article is a review that focuses on nutrition to treat asthma, an atopic disorder.  

Point 2: 

4.2 You should try to explain why the efficacy of the new therapy is significant  only for 30 minutes after nasal challenge. This could be an inportant limit in its future use in clinical practice. 

Response 2:

This is explained in the discussion: 

“Unfortunately, although the combination nasal corticosteroid combined with oral sul-foraphane intervention group showed improved TNSS when compared to placebo, this difference was not maintained outside of 30 minutes.  TNSS scores are lowest within the first hour following nasal challenge and gradually improve(31).  The combination of anti-oxidant SFN and nasal corticosteroid may decrease the severity of the early phase which involves release of mast cell mediators though may fail to prevent the effect of the late phase, which involves arrival of other inflammatory cells (32).”         

Page 14: line 402-410.   

   CONCLUSION

Point 1:

You should modify the first sentences. The subjective benefit of this therapy is not so evident!

Response 1:

This is corrected to “Our study shows objective benefits with the combination of nasal corticosteroids and sulforaphane when compared to placebo tablets and nasal saline, although measured in-flammatory markers were not affected and subjective improvement was only temporary”

REFERENCES

Point 1:  

Control orthograpy error and punctuation  in 7, 10, 15 24,26,28,32,37

Response 1:

These have been corrected

Reviewer 2 Report

Dear editors,
This manuscript is interesting as it discusses the application of BSE for treatment of allergic rhinitis. The setup of the study appears sound, and it would seem the data are adequately analyzed. However, the representation of the data can be improved upon. 

In the introduction, although the reading audience may well be able to put the storyline together, what appears to be lacking most is a well defined hypothesis which would tie the presented findings together. 
Furthermore, some parts require further explanation, e.g. "Atopic patients diagnosed with asthma are more likely to have a an impaired ability to produce an adequate amount of phase II enzymes since they are most likely to have glutathione transferase (GST) null genotypes (18) and these individuals tend to have an exacerbated T2 response to diesel exhaust (19)." - what does it mean when phase II enzyme-levels are lower, how does that tie into rhinitis, what IS an adequate level. If vitamin E and C therapy are representatives of antioxidant therapy, that should be indicated as such.

Another example is "Administration of SFN increased expression of phase II enzymes while inhibiting proinflammatory cytokines in mice bronchial epithelial cells after exposure to diesel exhaust particles (25)." what ARE those products?

Furthermore, the line "The primary objective of this study was to enroll males and females 18 years..." should be rephrased

the focus on the diesel exhaust particles is somewhat confusing; this isn't part of the current study

study design:
in general, the description of the design would benefit from a graphical representation of the experimental setup. furthermore, there appears to be some repetition in the subparagraphs (which is at times unavoidable, but in this manuscript overlap can be reduced)

2.2 "After discussion and thought, if interested, all participants provided written informed consent." should be rephrased

2.3 "will be derived" use only past tense

"biomarkers included IL-4,IL-1b and IL-13" were there more and are these not mentioned? I'd suggest not mentioning these here

2.4 "Subject were randomized to take BSE or Placebo 4 tablets each morning Nasal Corti-costeroid Spray: Fluticasone propionate nasal spray prescribed as 2 sprays in each nostril Placebo Nasal Spray containing was used as the placebo nasal spray." I am not a native speaker, but I really cannot make sense of this sentence.

Results:
the graphs could be improved upon, especially the layout of the axes. also, there is mixed use of abbreviations and full descriptions of the scores.

discussion:

the authors sate there is an overall significant improvement but there are several points at which the significance levels do not meet those criteria. 

I would like to see more explanation regarding the mechanism of action, and a more integrated discussion (instead of discussing the results per section, combine the results and draw conclusions/compare to other studies.

I find it difficult to piece together the different aspects of the outcomes (also because this isn't clearly hypothesis-driven)

Author Response

INTRODUCTION

Point 1:

In the introduction, although the reading audience may well be able to put the storyline together, what appears to be lacking most is a well defined hypothesis which would tie the presented findings together. 
Furthermore, some parts require further explanation, e.g. "Atopic patients diagnosed with asthma are more likely to have a an impaired ability to produce an adequate amount of phase II enzymes since they are most likely to have glutathione transferase (GST) null genotypes (18) and these individuals tend to have an exacerbated T2 response to diesel exhaust (19)." - what does it mean when phase II enzyme-levels are lower, how does that tie into rhinitis, what IS an adequate level. If vitamin E and C therapy are representatives of antioxidant therapy, that should be indicated as such.

Response 1: 

Phase II enzymes is explained they act as antioxidants in the revised introduction.   A decrease in Phase II enzymes leads to increase oxidation that may lead to increase symptoms of allergic rhinitis.  Phase II is broad term for enzymes required for antioxidation.   It has been added that both vitamin E and C are considered antioxidants.  

Page 1: line 74-75.

Point 2: 

Another example is "Administration of SFN increased expression of phase II enzymes while inhibiting proinflammatory cytokines in mice bronchial epithelial cells after exposure to diesel exhaust particles (25)." what ARE those products?

Response 2:

SFN increases Phase II enzyme while blocking IL-8, GM csf and IL1beta production.   This has been added to the paragraph       

Page 1: line 79-80

Point 3: 

Furthermore, the line "The primary objective of this study was to enroll males and females 18 years..." should be rephrased

 Response 3:

This has been addressed in revisions taken to explain diesel exhaust particle exposure leads to increase in oxidation.  Since this is mentioned later in the paper, it was removed.      

Page 1: line 89-90

Point 4: 

the focus on the diesel exhaust particles is somewhat confusing; this isn't part of the current study

Response 4: 

Diesel Exhaust was the pollutant that was mentioned to emphasize the role of oxidation in allergy and the need for supplementation with antioxidants.   Work utilizing diesel exhaust was chosen as the pollutant since it has been studied in our investigator’s lab.  

METHODS:

Point 1:

In general, the description of the design would benefit from a graphical representation of the experimental setup. furthermore, there appears to be some repetition in the subparagraphs (which is at times unavoidable, but in this manuscript overlap can be reduced)

Response 1:

Graphics of the study protocol was added as figure 1

Page 6: line 243.  

Point 2:

2.2 "After discussion and thought, if interested, all participants provided written informed consent." should be rephrased

Response 2:

This sentence was removed, since consent is mentioned in following paragraph.   

Page 3: line 121-122.  

Point 3: 

2.3 "will be derived" use only past tense

Response 3: 

Will be derived was omitted and replaced with sentence: Clinical measurements were obtained using the total nasal symptoms score (TNSS) and peak nasal inspiratory flow (PNIF).    

Page 3: line 135. 

Point 4:

"biomarkers included IL-4,IL-1b and IL-13" were there more and are these not mentioned? I'd suggest not mentioning these here

Response 4:

In the following sentence the list of all biomarkers (cytokines) measured is listed:  During nasal challenges, the following cytokines were measured: IL4, IL5, IL13,IL6, IL8 and IL1b

Page 3: line 138-139

Point 5:  

2.4 "Subject were randomized to take BSE or Placebo 4 tablets each morning Nasal Corti-costeroid Spray: Fluticasone propionate nasal spray prescribed as 2 sprays in each nostril Placebo Nasal Spray containing was used as the placebo nasal spray." I am not a native speaker, but I really cannot make sense of this sentence

Response 5:  

The sentence above was omitted and replaced with the following sentences:

Following baseline nasal challenge, subjects were randomized into one of 4 groups: 

.1.nasal corticosteroid + Placebo tablet

  1. placebo nasal spray + Placebo tablet
  2. nasal corticosteroid + Avmacol® (SFN supplement)
  3. placebo nasal spray + Avmacol® (SFN supplement)

Avmacol is a supplement that supports sulforaphane production by providing glu-coraphanin and an active myrosinase enzyme via our Sulforaphane Production System.  Subjects were given 4 tablets to take with their evening meal. 

Page 4: line 154-157.  

RESULTS:

Point 1: 

The graphs could be improved upon, especially the layout of the axes. also, there is mixed use of abbreviations and full descriptions of the scores

Response 1: 

Graphs were redone, addressing the issues above.        

DISCUSSION:

Point 1:  

The authors state there is an overall significant improvement but there are several points at which the significance levels do not meet those criteria.

I would like to see more explanation regarding the mechanism of action, and a more integrated discussion (instead of discussing the results per section, combine the results and draw conclusions/compare to other studies.

I find it difficult to piece together the different aspects of the outcomes (also because this isn't clearly hypothesis-driven)

Response 1:

Discussion was rewritten addressing the above concerns.  

Round 2

Reviewer 2 Report

Dear authors,

The 'lack' of a hypothesis has been sufficiently addressed by rewriting the first part of the introduction. The explanation for phase II enzymes is also clear.
Addition of the figure explaining the study setup has improved the legibility of the main text, in my opinion. Other minor issues related to the text have been improved sufficiently, but careful re-reading is still required (e.g. line 339 now reads "Although various investigations have investigated...").

I do still feel there is a lot of emphasis on the diesel exhaust fumes and although I understand that is is given as an example of a pollutant causing oxidation, when reading the introduction, I would (as a reader) expect that DEP would be part of the study setup, which it isn't.

It doesn't need to be excluded but I would stress the 'importance' of your own study to a greater extent; i.e. make sure that the readers don't think that looking at the effects of BSE without DEP (as an additional cause for oxidative stress) is still a valid approach. 

The discussion had been restructured somewhat and the addition of the text in lines 362-367 addresses my earlier remark about the mechanism of action. 

Author Response

Response to Reviewer 2 Comments

Point 1:   Other minor issues related to the text have been improved sufficiently, but careful re-reading is still required (e.g. line 339 now reads "Although various investigations have investigated...").

Response 1:     First sentence in discussion was changed to “Recent investigations indicate a role of antioxidants/supplements in the treatment of asthma and allergic rhinitis.”   In addition, references in the discussion were corrected to include 29,30 (prior was lumped together) and one article reference was replaced with the original article ((reference 35 omitted and reference 38 was added).       

Page 10: line 319-320    

Page 16: line 492

Page 17: line 506-507, then line 513-515. 

 Point 2:  I do still feel there is a lot of emphasis on the diesel exhaust fumes and although I understand that is is given as an example of a pollutant causing oxidation, when reading the introduction, I would (as a reader) expect that DEP would be part of the study setup, which it isn't.

It doesn't need to be excluded but I would stress the 'importance' of your own study to a greater extent; i.e. make sure that the readers don't think that looking at the effects of BSE without DEP (as an additional cause for oxidative stress) is still a valid approach.

Response 2:  The following statement was added to stress the importance of the effect may only occur when there is significant oxidant exposure, such as with DEP.      

“These findings may indicate a role of antioxidant supplementation in treating allergic rhinitis in high pollutant areas.”

Page 12: line 393-394